# Ecological Niche Studies on *Hylurgus ligniperda* and Its Co-Host Stem-Boring Insects

Lihong Bi [1], Jing Tao [1,*], Lili Ren [2], Chuanzhen Wang [2] and Kai Zhong [2]

1   Beijing Key Laboratory for Forest Pest Control, Beijing Forestry University, Beijing 100083, China; bilihong@bjfu.edu.cn
2   Yantai Forest Resources Monitoring and Protection Service Center, Yantai 264000, China; lily_ren@bjfu.edu.cn (L.R.); wczhyt@163.com (C.W.); zhongkai@yt.shandong.cn (K.Z.)
*   Correspondence: taojing1029@hotmail.comtaojing@bjfu.edu.cn

**Abstract:** *Hylurgus ligniperda* (Fabricius), a significant quarantine pest, has recently invaded China, marking a new spread outside its known global distribution. This study aims to clarify the invasion and colonization mechanisms of *H. ligniperda* in Shandong Province, a primary colonization site. This study employed sampling surveys and analysis of damaged wood, discovering that the wood-boring insects sharing the same host as *H. ligniperda* mainly include *Cryphalus* sp., *Arhopalus rusticus*, and *Shirahoshizo* sp. Through ecological niche theory, the study analyzed the ecological niche relationships between *H. ligniperda* and these three wood-boring insects, from the perspectives of temporal and spatial resource utilization. The results reveal that these insects could cause damage to *P. thunbergii* trees at different health levels, with *H. ligniperda* being the most destructive. The ecological niches of insect populations varied significantly by tree vigor and height. *Cryphalus* sp. occupied the entire trunk, whereas *A. rusticus* and *Shirahoshizo* sp. were concentrated in the lower-middle trunk and the root section up to a depth of 1 m. Notably, *H. ligniperda* primarily targeted tree roots. Due to the differences in spatial distribution, there was no intense competition between *H. ligniperda* and other wood-boring insects. With a decline in the health of the host tree, *Cryphalus* sp. ascended the trunk, whereas *H. ligniperda* spread deeper into the roots and *A. rusticus* moved towards the base of the trunk and the top of the roots. *Shirahoshizo* sp. showed a less defined distribution pattern. Therefore, *H. ligniperda* was more dominant during the later stage of damage. The position occupied by each insect on the trunk was relatively stable, and the ecological niche overlap value with *H. ligniperda* was low in terms of temporal resources. Therefore, *H. ligniperda* and other stem-boring pests exhibit coexisting populations mainly through the allocation and utilization of spatial resources, eventually promoting the successful colonization of *H. ligniperda*.

**Keywords:** *Hylurgus ligniperda*; spatial and temporal ecological niche; boring insect; interspecific relation





## 1. Introduction

*Hylurgus ligniperda* (J. C. Fabricius, 1787) belongs to the genus *Hylurgus* Latreille, tribe Tomicini, subfamily Scolytinae, family Curculionidae, and order Coleoptera. It is native to Europe, but is now distributed across all continents and classified as an internationally significant quarantine pest of forestry [1]. In China, *H. ligniperda* was first found in October 2020 in the protected coastal forests of Yantai City and Weihai City in Shandong Province.

*H. ligniperda* develops large populations with overlapping generations, with one generation per year in France and three generations per year in Chile. Thus, it can cause damage throughout the year. The peak flight activity of adults after emergence generally occurs during mid-spring and late summer into fall [2,3]. *H. ligniperda* mainly occupies dead or fallen pine trees in Europe, thus belonging to the category of secondary pests [4,5]. However, it has a strong diffusion ability, as its invasion and colonization have been reported from several countries or regions, including Australia, Japan, New Zealand, South

Africa, parts of South America (Argentina, Brazil, Chile, Paraguay, and Uruguay), Sri Lanka, the United States (New York and California), Korea, and China [6–9], and it is believed to have the fastest diffusion speed [10].

Since its spread in China, *H. ligniperda* has caused great harm to the protected coastal forest area in Yantai, Shandong Province. *H. ligniperda* can affect trees with suboptimal health, and its adults can invade the roots of pine trees directly from the surface of the trunk, feeding on the trunk and the root phloem [8]. *H. ligniperda* has a strong reproductive ability, a large population, and significant generation overlapping. Adults lay eggs in root cavities, and when the eggs hatch, they feed on the root bast together with the larvae, which can destroy the entire root. *H. ligniperda* and other boring insects can jointly harm the same pine tree, thereby accelerating the death of the tree.

Studies have projected the potential geographic distribution of *H. ligniperda* globally under recent and future climatic scenarios and reported that the Mediterranean periphery, the eastern seaboard of Asia, and southeastern Oceania are highly conducive to its spread [11]. Moreover, in China, *H. ligniperda* occupies a wide range of habitats, including nearly all provinces of central and southern China [12,13]. *H. ligniperda* exhibits high tolerance to extreme temperatures during different developmental stages [14]. It can carry pathogens such as the blue stain fungi *Ophiostomatales*, which infest and harm host trees, affecting wood quality [15,16]. Thus, *H. ligniperda* has a strong potential to harm forests.

Previous research on *H. ligniperda* was oriented towards investigating the compositions and types of associated fungal populations and related bacterial communities, to explore the roles they may play in the invasion process of *H. ligniperda* [17,18], its life cycle [19,20], and its detection and trapping effectiveness [21]. However, *H. ligniperda* can damage the host *Pinus thunbergii* (Parl) alongside other native pests in newly invaded areas. Whether there is competition between them in terms of the utilization of temporal and spatial resources, and how they achieve coexistence, remains unknown. Therefore, utilizing niche theory to explore the relationships between *H. ligniperda* and other stem-boring insects sharing the same host is particularly important in order to clarify the invasion mechanism of *H. ligniperda*.

Ecological niche theory is among the important elements of modern ecological research and has been widely used since the concept of the ecological niche was first proposed. Many scholars have conducted studies on ecological niche theory, mainly reflecting the relationships between populations in the ecosystem, including the allocation and utilization of natural resources; competition and coexistence between species; the statuses and roles of organisms in the environment; and the stability of the ecosystem [22–26].

During a study on bark beetles, Chen Hui et al. conducted research on the species and ecological niches of pine bark beetles, identifying that *Dendroctonus armandi* (Tsai & Li, 1959) is a pioneer species. It utilizes the remaining nutrients and space of its host, *Pinus armandii* (Franch), thus achieving dynamic stability in the ecosystem of standing *P. armandii* bark beetles in the Qinling Mountains [27]. Liu Li et al. applied niche theory in order to study the spatial niches of bark beetle populations in natural forests of *Picea crassifolia* (Kom), clarifying that the diversity in the selection and utilization of spatial resources by bark beetle populations has led to a balance and coexistence on *P. crassifolia* [28]. Yuan Fei et al. studied the spatial ecological niches of the main stem-boring pests of *Larix gmelinii* (Rupr) in the Aershan area. The results showed that the ecological niche of *Ips subelongatus* (Bright & Skidmore, 2002) was highest on weakened standing trees. Although the interspecific spatial competition among the pests was intense, coexistence was achieved through the differentiation of feeding sites [29]. It can be seen that by integrating niche theory, the interrelationships among species—such as competition, coexistence, and resource utilization—can be clarified. This further elucidates the roles of organisms in the environment and in the stability of ecosystems. Because of the above, we conducted a study on the ecological niches of *H. ligniperda* and other stem-boring insects sharing the same host, in order to clarify the invasion mechanism of *H. ligniperda*.

Given the recent emergence of *H. ligniperda* as a novel invasive species in China, its ecological adaptability and coexistence mechanisms urgently need further research. Our survey found that the adults and larvae of *H. ligniperda*, the adults and larvae of *Cryphalus* (W.F.Erichson, 1836), the larvae of *Arhopalus rusticus* (Linnaeus, 1758), and the larvae of *Shirahoshizo* (K. Morimoto, 1962) can coexist and cause harm in the same host tree. However, how these insects cleverly utilize temporal or spatial resources to achieve coexistence remains unknown. Whether there is a competitive relationship between these insects and how this potential competition affects their population dynamics and distribution patterns are also issues worthy of further exploration. This is also the reason why we conducted this study. Based on ecological niche theory, the study was divided into two dimensions—temporal and spatial—to reveal the coexistence mechanism and interaction relationship between these insects; this will help us to better understand the ecological adaptation strategies of invasive species, providing useful references for the prevention and management of other similar situations.

## 2. Materials and Methods

### 2.1. Overview of the Experimental Site

The study site was located in the protected coastal forest of Muping District, Yantai City, Shandong Province (37.46° N, 121.85° E). This forest belongs to the temperate monsoon climate and is mainly dominated by *P. thunbergii* trees, a species introduced through plantation.

### 2.2. Research Subjects and Sample Collection

This experiment aimed to explore the variations in the distribution of insect populations in trees of different health levels and heights. To this end, a total of 18 representative host *P. thunbergii* trees, which were affected by *H. ligniperda*, were investigated as the subjects of the study. The average age of the sample trees was approximately 55 years, and each had a mean height of 9.7 m and a mean diameter at breast height (DBH) of 17.4 cm (Table 1).

**Table 1.** Measurements of the Sample Trees.

|  | Quantity | Mean Height (m) | Mean DBH (cm) |
|---|---|---|---|
| Yellow–green tree | 6 | 9.92 ± 1.32 | 16.73 ± 2.57 |
| Red-crowned tree | 6 | 9.83 ± 1.63 | 18.87 ± 3.54 |
| Dead tree | 6 | 9.42 ± 1.86 | 16.52 ± 3.53 |

The 18 sample trees were felled at different times. In August, October, and December of 2022, and February, April, and June of 2023, three *P. thunbergii* trees of different health levels were randomly selected each month from the experimental site. The health level was indicated through each tree's appearance, with varieties including yellow–green trees, red-crowned trees, and dying trees; one of each type was selected for the collection of insect samples. During the collection process, we ensured the diversity and representative of the samples to obtain accurate research results.

Characteristics of Insect Species

Boring habits: *H. ligniperda* mainly attacks the base and root of the trunk, primarily feeding on the phloem. *Cryphalus* sp. mainly feeds on the phloem under the bark of the trunk [30]. The newly hatched larvae of *A. rusticus* feed under the bark. After 4 to 6 weeks, they bore into the phloem and cambium to feed, and then gnaw the xylem inward [31]. Larvae of *Shirahoshizo* sp. drill into the bark layer of the host [32]. The adults and larvae of *H. ligniperda* and *Cryphalus* sp. and the larvae of *A. rusticus* and *Shirahoshizo* sp. at different stages of development can infest the same host tree. To further explore the temporal and spatial changes during the mixed infestation of these insects within the trunk, it is assumed that the damage caused by adults and larvae to the host tree is similar, as they all feed on the phloem. Therefore, adults and larvae at different stages of development are no longer distinguished and are counted together, providing a more accurate reflection of the infestation situation of these insects within the trunk.

Insect collection method: Direct excavation of the affected trees was conducted, and the experimental insects were obtained by meticulously dissecting the main root, trunk, lateral roots, and branches; next, the numbers of *H. ligniperda* and other stem-boring insects at various life stages, including adults, larvae, and pupae, were counted. Insects accidentally damaged during the dissection process were also counted. For ease of counting, larvae of these four types of insects were not differentiated by age and were counted together as a unified group.

Identification method: The species of the collected insects, including adults, pupae, larvae, and other developmental stages, were identified based on their morphological characteristics [33–36]. Larvae were further identified using molecular identification methods as supplementary verification.

### 2.3. Selection of Predictive Variables

To analyze the relationship between the distribution of insect populations and tree health and height, we selected the following predictive variables:

Tree vigor: *P. thunbergii* trees were divided into three categories based on changes in the color of their crown needles and symptoms of damage after invasion by the borer—namely, yellow–green trees (early stage of invasion), red-crowned trees (middle stage of invasion), and dead trees (late stage of invasion) [37].

Yellow–green trees had yellowish crown needles and some healthy green needles, no obvious entry and exit holes in the trunk, and fresh reddish-brown insect droppings that could be observed at the base of the trunk. Red-crowned trees had an overall yellowish crown with partly shed needles, trunks showed entry and exit holes, and several dried insect droppings were observed at the base of each trunk alongside fresh droppings. Dead trees had an overall reddish crown, dry needles, several entry and exit holes, and older dried droppings at the base of their trunk (Figure 1).

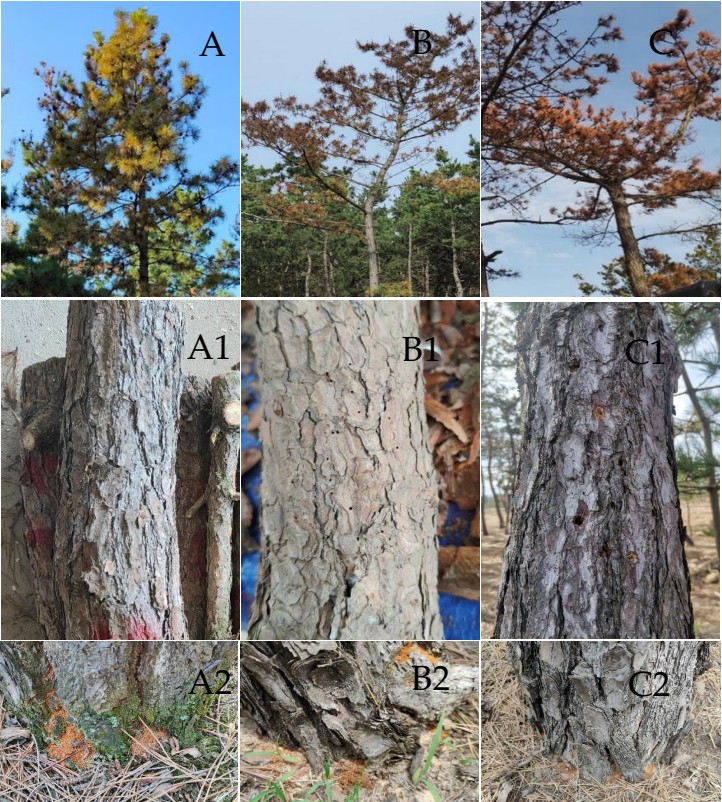

**Figure 1.** Health status of *Pinus thunbergii* after boring insect damage: (**A**) tree with yellow–green foliage; (**B**) tree with red-crowned appearance; (**C**) dead tree; (**A1–C1**) entrance and emergence holes on the trunk surface; (**A2**) fresh frass discharged at the base of a trunk; (**B2**) fresh and old frass discharged at the base of a trunk; (**C2**) old frass discharged at the base of a trunk.

Height: Based on the height range of the trees, each tree was segmented by height at 1 m intervals. Preliminary examination revealed no pest damage to the trunk above 9 m in the yellow–green tree, whereas in the red-crowned and dead trees, the trunk above 9 m had dried up, and entrance and emergence holes could be observed but no boring insects were found. For the convenience of the survey, heights of 1 m, 3 m, 5 m, 7 m, and 9 m were selected, along with heights 1 m and 2 m below ground level for the root part which were labeled as −1 m and −2 m, respectively.

### 2.4. Statistical Chart of the Quantity Ratio of Various Boring Insects under Different Tree Vigors

Data preparation: classify the collected insect samples according to their tree vigor.

Within the tree vigor units, the same tree vigor is considered as one category, with a total of 3 categories, and each category has 6 samples. Calculate the total number of all boring insects under the same tree vigor, and then count the number of each type of boring insect separately. Calculate the quantity ratio of each insect's number under the same tree vigor.

Chart construction: use the quantity ratio of insects under the same tree vigor as the dependent variable, and use tree vigor as the independent variable.

### 2.5. Statistical Chart of Average Insect Population Density at Different Heights under the Same Tree Vigor

Data preparation: classify the collected insect samples according to their heights.

Calculate the total number of individuals for each insect species at different heights of yellow–green trees, red-crowned trees, and dying trees, respectively. Then calculate the average number of insects under the same tree vigor and the same height of each tree.

Chart construction: use the average number of insects under the same tree vigor and the same height of each tree as the dependent variable, and the tree condition as the independent variable.

### 2.6. Generalized Linear Model (GLM) Analysis

To analyze the variation in insect population distribution under different tree vigors and heights, we employed a Generalized Linear Model (GLM) for statistical analysis.

Data preparation: the collected insect samples of different species were categorized according to tree vigor and height.

For the height unit, all wood segments at the same height formed one group, totaling seven groups, with each group comprising eighteen segments.

For the tree vigor unit, samples were classified based on the overall health status of the whole plant, with plants of the same health status forming one category, totaling three categories, with each category having six trees. Under yellow–green trees, red-crowned trees, and dying trees, each health status category has six trees, and each tree has seven different heights. Each tree forms one group based on all wood segments at the same height. There are seven different heights under each health status, divided into seven groups, with each group having six segments of wood at the same height. The number of different insects in each group was then counted separately.

Model construction: a GLM model was constructed with the number of insects as the dependent variable, and tree vigor and height as independent variables.

Model fitting and testing: Statistical software was used to fit the model, and omnibus tests were conducted to determine whether the dependent variable in the model was significantly affected by one or more independent variables. The applicability and accuracy of the model were tested using methods such as tests for model effects.

### 2.7. Temporal Niche Analysis

According to the survey times in August, October, and December 2022, and February, April, and June 2023, the number of various boring insects collected each month was counted and the temporal niche overlap index of each insect was calculated, respectively.

*2.8. Niche Value Calculation Formula*

2.8.1. Niche Width

Niche width was calculated using the following formula proposed by Levins (1968):

$$B = 1/(s \sum_{i=1}^{s} P_i^2) \tag{1}$$

where B represents the species' niche breadth and R represents the number of available resource states. $P_i$ is the proportion of species in unit i.

2.8.2. Ecological Niche Overlap Index Was Calculated Using the Equation

Equation (2):

$$a_{ij} = \sum_{h=1}^{n} P_{ih} P_{jh} (B_i) \tag{2}$$

where $a_{ij}$ is the ecological niche overlap of species i over species j; $P_{ih}$ and $P_{jh}$ are the proportions of species i and j, respectively, in unit h of the resource set; and $B_i$ is the ecological niche width of species i.

2.8.3. The Ecological Niche Similarity Coefficient (PS) Was Calculated Using the Equation

Equation (3):

$$PS = 1 - \frac{1}{2} \sum_{i=1}^{n} |P_{ij} - P_{hj}| \tag{3}$$

where $P_{ij}$ and $P_{hj}$ are the proportions of species i and h in the resource unit j.

2.8.4. Coefficient of Ecological Niche Competition

The interspecific competition was measured using May's (1975) coefficient of interspecific competition ($\alpha$):

$$\alpha = \sum P_i P_j / (\sqrt{\sum P_i^2} \sqrt{\sum P_j^2}) \tag{4}$$

where $\alpha$ is the coefficient of competition between species i and species j in the same resource, while $P_i$ and $P_j$ denote the proportions of species i and j in each resource sequence, respectively.

## 3. Results

*3.1. Species Abundance and Distribution of Boring Insects in P. thunbergii Trees*

Over the course of a year, surveys during every alternate month revealed that the main drilling insects were *H. ligniperda*, *Cryphalus* sp., *A. rusticus*, and *Shirahoshizo* sp., all of which occupied different positions on *P. thunbergii* trees, and their abundance and distribution varied with different survey times and progression of damage.

As shown in Figure 2, *H. ligniperda*, *Cryphalus* sp., *A. rusticus*, and *Shirahoshizo* sp. can harm *P. thunbergii* trees by varying degrees. Combined with data from Figures 3–5, *Cryphalus* sp. was observed to mainly affect the trunk and not the roots, although due to its large population, it could harm the entire trunk. Moreover, its distribution in the yellow–green and red-crowned trees was as high as 82.89% and 63.113%, respectively, and it shifted to the higher part of the trunk as the tree weakened, with a high degree of dispersion at different heights.

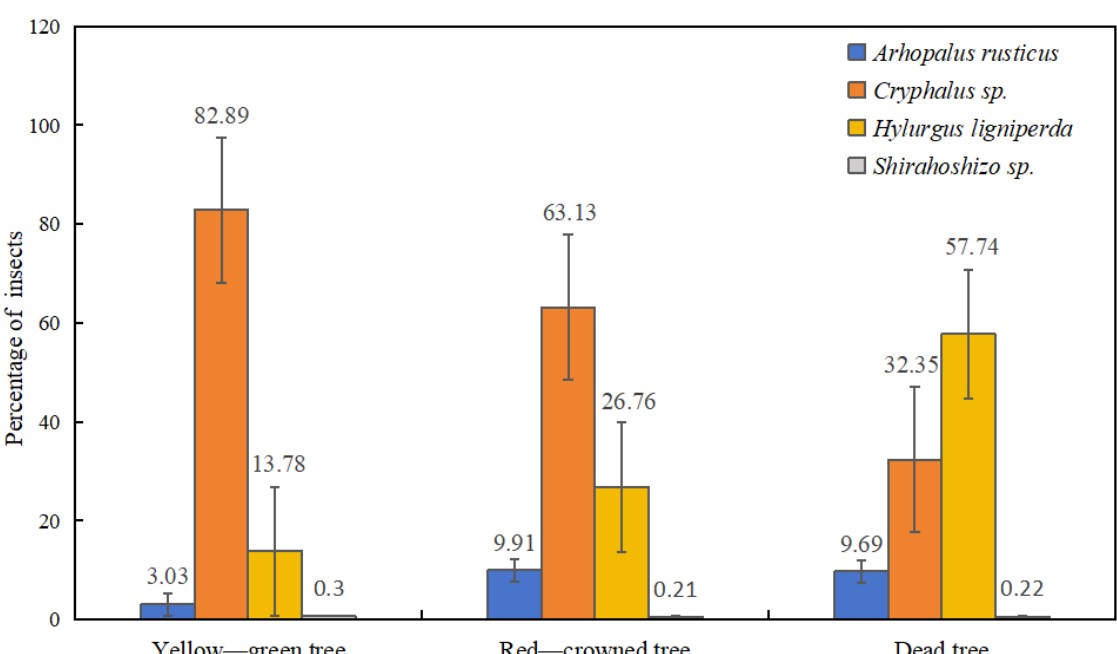

**Figure 2.** Percentages of the four boring insect species detected after bark removal from *Pinus thunbergii*, categorized into three health statuses. Note: the error bars in the figure represent the standard deviation.

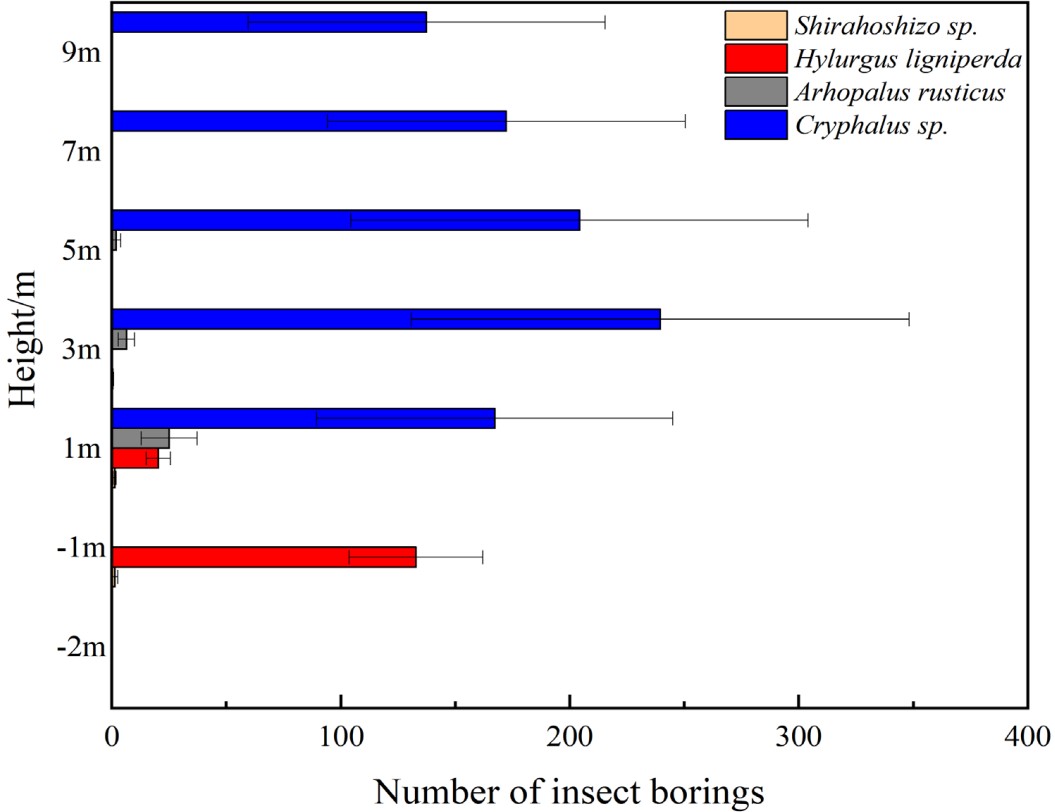

**Figure 3.** Average distribution map of insect numbers at different heights (−1~9m) on six yellow–green trees. Note: The error bars in the figure represent the standard deviation. "−1 m" represents a height of 1 m from the base of the trunk to the root, while "−2 m" denotes a 1 m long log section cut at a position 2 m from the base of the trunk towards the root.

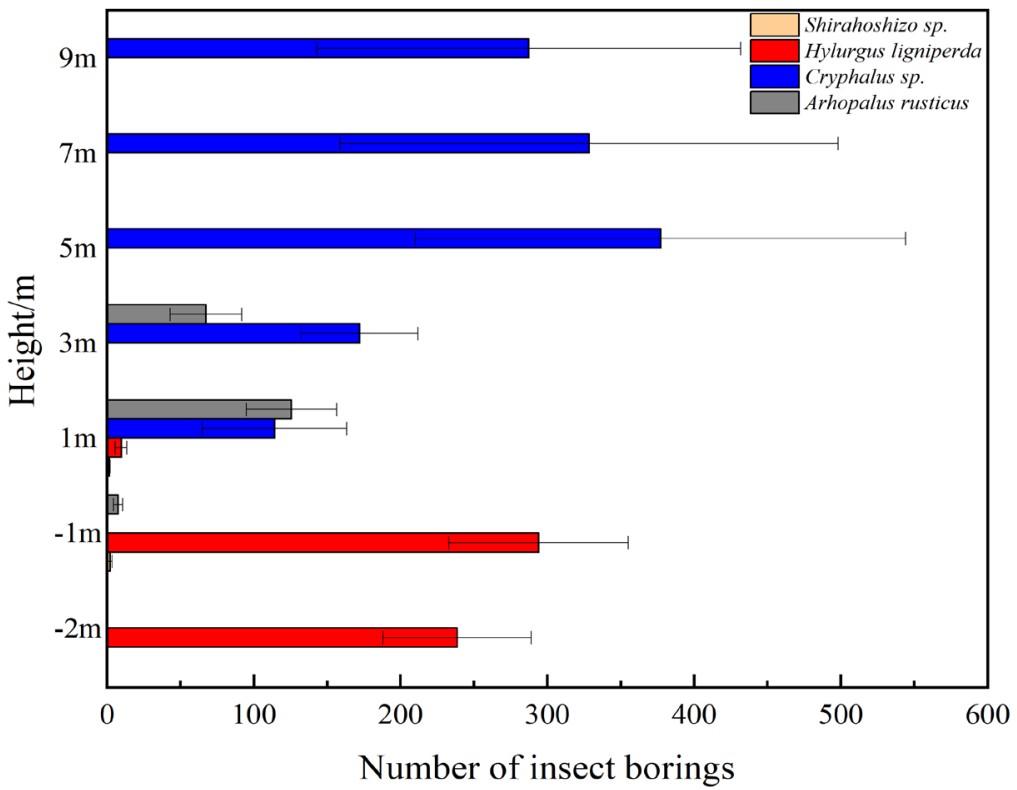

**Figure 4.** Average distribution map of insect numbers at different heights (−1~9m) on six red-crowned trees.

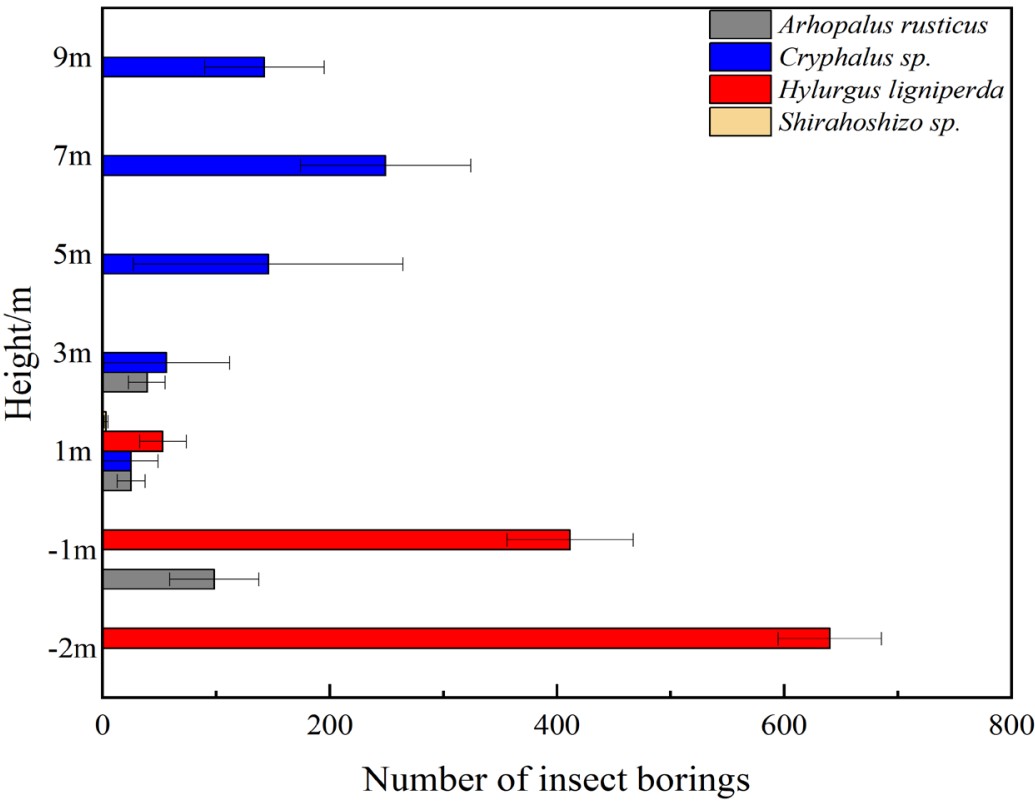

**Figure 5.** Average distribution map of insect numbers at different heights (−1~9 m) on six dead trees.

The niche occupied by *H. ligniperda* was from the base of the trunk to the entire root, and its population increased with the weakening of the host tree. Its distribution in the dead tree was 57.74%, with dispersion deeper into the root with the weakening of the tree.

*A. rusticus* occupied a niche from 1 m at the root to 5 m at the trunk, mainly distributed at the base of the trunk. With the weakening of the tree, its distribution spread to the roots and the 3 m section of the trunk (Figures 3–5). *Shirahoshizo* sp. infested the roots and the 1–3 m sections of the trunk, mostly concentrated at the base of the trunk. As the overall population of *Shirahoshizo* sp. was relatively small (up to a few dozen), its proportion of distribution in *P. thunbergii* trees at different degrees of health was only 0.2–0.3%.

As shown in Table 2, the *p*-value in the Omnibus test is less than 0.05, indicating that there are significant differences between the independent variables and the dependent variable. This means that height, tree vigor, and the interaction between tree vigor and height have significant differences in the distribution proportions of stem-boring insects (Table 2).

**Table 2.** Analysis of the comprehensive impacts of *Pinus thunbergii* tree vigor and height on the numbers of four kinds of insects: omnibus test results, $p < [0.05]$.

| | Omnibus Test | | |
|---|---|---|---|
| | LR $\chi^2$ | Df | Sig |
| *H. ligniperda* | 289.166 | 20 | 0.000 |
| *Cryphalus* sp. | 149.551 | 20 | 0.000 |
| *A. rusticus* | 106.408 | 20 | <0.001 |
| *Shirahoshizo* sp. | 67.540 | 20 | <0.001 |

The results of the test of model effects indicate that the tree strength and height of *P. thunbergii* trees, as well as the interaction between these factors, have a significant impact on the distribution and the number of the three insects, *H. ligniperda*, *Cryphalus* sp., and *A. rusticus* ($p < 0.05$). There is no significant difference in the number of *Shirahoshizo* sp. under different tree strengths (Tables 2 and 3).

**Table 3.** Analysis of the comprehensive impacts of *Pinus thunbergii* tree vigor and height on the numbers of four kinds of insects: test of model effects results $p < [0.05]$.

| | Test of Model Effects | | | |
|---|---|---|---|---|
| | Source | Wald $\chi^2$ | Df | Sig |
| *H. ligniperda* | Intercept | 296.365 | 1 | 0.000 |
| | Tree vigor | 124.614 | 2 | 0.000 |
| | Height | 651.848 | 6 | 0.000 |
| | Tree vigor × Height | 347.974 | 12 | 0.000 |
| *Cryphalus* sp. | Intercept | 387.698 | 1 | 0.000 |
| | Tree vigor | 31.966 | 2 | <0.001 |
| | Height | 201.794 | 6 | 0.000 |
| | Tree vigor × Height | 53.136 | 12 | <0.001 |
| *A. rusticus* | Intercept | 70.116 | 1 | <0.001 |
| | Tree Vigor | 20.364 | 2 | <0.001 |
| | Height | 106.173 | 6 | 0.000 |
| | Tree vigor × Height | 40.643 | 12 | <0.001 |
| *Shirahoshizo* sp. | Intercept | 35.458 | 1 | <0.001 |
| | Tree vigor | 0.436 | 2 | 0.804 |
| | Height | 71.043 | 6 | <0.001 |
| | Tree vigor × Height | 17.880 | 12 | 0.119 |

The overall ecological niches of these insects in the tree host are presented in Figure 6.

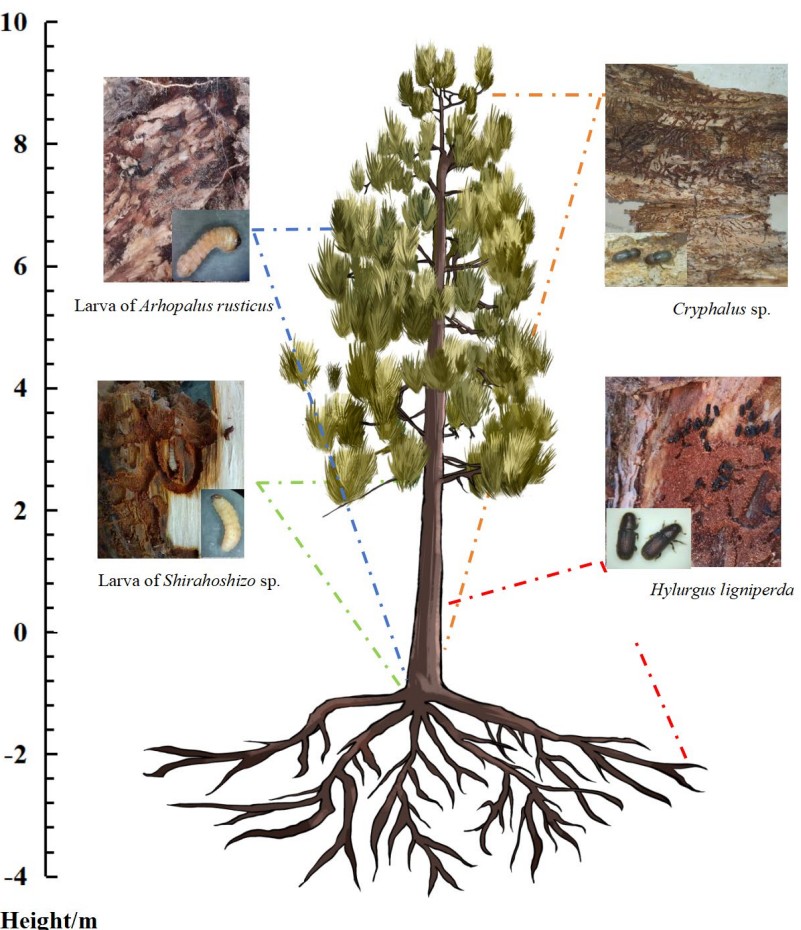

**Height/m**

**Figure 6.** Ecological niches of the four insect species in *Pinus thunbergia* trees.

*3.2. Temporal Ecological Niche*

Temporal niche refers to the ecological niche of a species in the time dimension, which describes the pattern of a species' activities on a specific time scale.

When *H. ligniperda* and several other insects feed and cause harm together during different survey periods, they overlap in temporal niches (Figure 7).

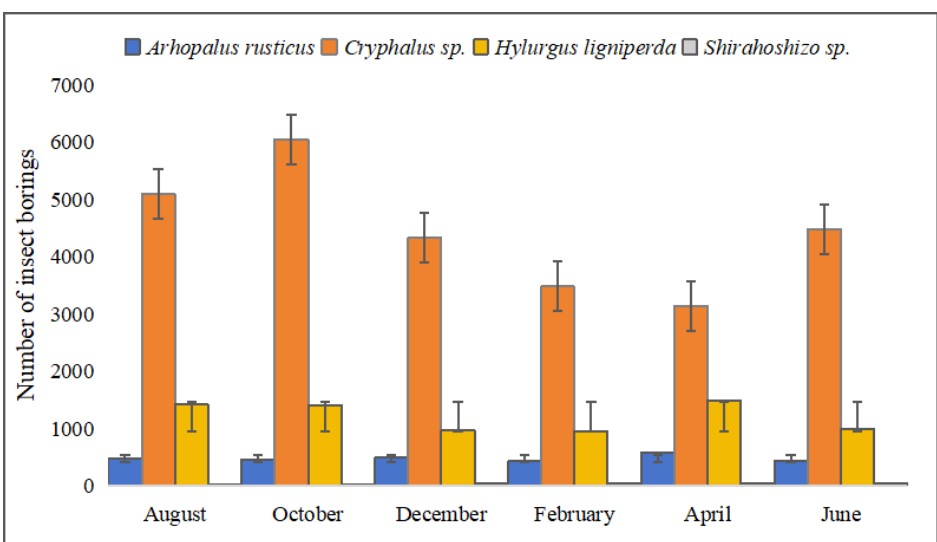

**Figure 7.** The number of boring insects in different survey periods. Note: the error bars in the figure represent the standard deviation.

The degree of temporal niche overlap between *H. ligniperda* and *Cryphalus* sp. is the highest (0.733), indicating a high overlap in the utilization of temporal resources, while the degree of temporal niche overlap with *Shirahoshizo* sp. is the lowest (0.008), which is conducive to the coexistence of the two species. The overlap degree with *A. rusticus* is moderate, and there is a certain overlap in temporal niches, which may lead to a certain degree of competition (Table 4).

**Table 4.** The overlap index of ecological niches among insects.

| | *Hylurgus ligniperda* | *Cryphalus* **sp.** | *Shirahoshizo* **sp.** | *Arhopalus rusticus* |
|---|---|---|---|---|
| *Hylurgus ligniperda* | | 0.733 | 0.008 | 0.289 |
| *Cryphalus* sp. | 0.733 | | 0.005 | 0.267 |
| *Shirahoshizo* sp. | 0.008 | 0.005 | | 0.014 |
| *Arhopalus rusticus* | 0.289 | 0.267 | 0.014 | |

### 3.3. Spatial Ecological Niche

Using the degree of health of *P. thunbergii* trees at the time of harvest as a resource sequence, the ecological niche widths of each boring insect in *P. thunbergii* trees at different health levels were analyzed.

From the findings summarized in Figure 8, the ecological niche width values of *Cryphalus* sp. were the highest in all three health states of *P. thunbergii* trees (0.8500, 0.8698, and 0.8437 in yellow–green trees, red-crowned trees, and dead trees, respectively). This was followed by *A. rusticus*, with ecological niche values of 0.6938, 0.7439, and 0.7212, respectively. The ecological niche width values of *H. ligniperda* and *Shirahoshizo* sp. were lower than those of the other insects. However, the ecological niche width values of *H. ligniperda* were higher in red-crowned and dead trees than in yellow–green trees, mainly because its distribution shifted deeper into the roots in dead trees. Overall, there was little variation in the spatial positions occupied by each of the four insect species in *P. thunbergii* trees of different health levels.

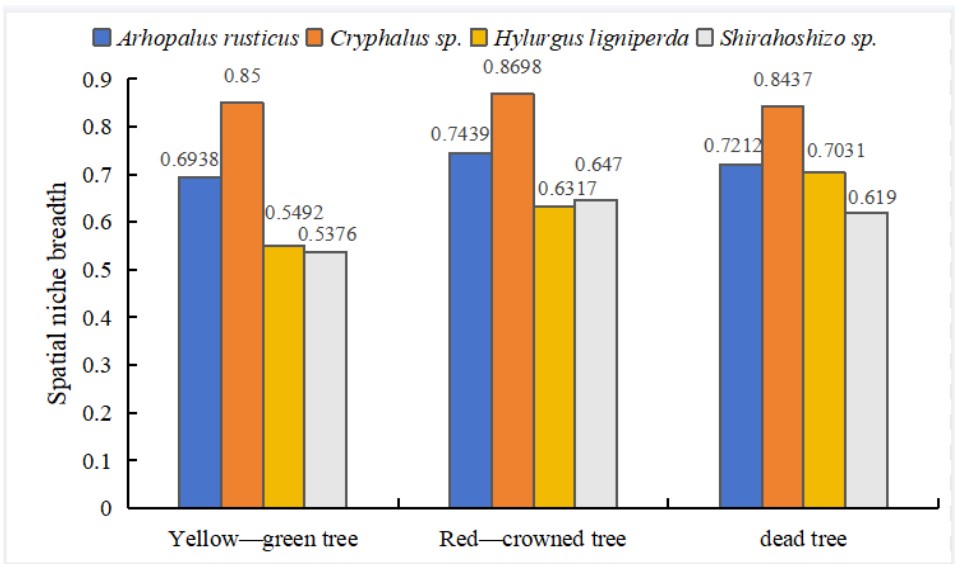

**Figure 8.** Ecological niche widths of various insects on *Pinus thunbergii* trees at different health levels.

The ecological niche overlap values between *H. ligniperda* and the other three insects in *P. thunbergii* trees at different health levels were the lowest (0.1118, 0.1079, and 0.0683), while those between *Cryphalus* sp. and *A. rusticus* were the highest (0.1565, 0.1606, and 0.1558). This is mainly because *H. ligniperda* was only distributed up to 1 m on the trunk, where the distribution of resources was low. However, *Cryphalus* sp. was distributed all over the trunk, while *A. rusticus* was mainly distributed across the middle and lower trunk, resulting in a higher degree of resource sharing between the two.

The maximum ecological niche similarity in *P. thunbergii* trees at different health levels was observed between *Cryphalus* sp. and *A. rusticus* (0.6834, 0.6567, and 0.5903), while the minimum was found between *Cryphalus* sp. and *Shirahoshizo* sp., followed by *H. ligniperda*. This is because *H. ligniperda* shares resources in only a 1 m section of the trunk, and although *Shirahoshizo* sp. can be distributed between 1 m and 3 m, the degree of resource utilization of *H. ligniperda* is much higher than that of *Shirahoshizo* sp., which explains the low ecological niche similarity proportion index between *Cryphalus* sp. and *Shirahoshizo* sp. (Table 5).

**Table 5.** Ecological niche overlap and similarity ratio among boring insects.

| Health Status of Pine Tree | Species | Niche Overlap (Proportion of Niche Similarity) | | |
|---|---|---|---|---|
| | | *Cryphalus* sp. | *A. rusticus* | *H. ligniperda* |
| | *A. rusticus* | 0.1565 (0.6834) | | |
| Yellow–green tree | *H. ligniperda* | 0.1118 (0.6111) | 0.1210 (0.5348) | |
| | *Shirahoshizo* sp. | 0.1542 (0.3987) | 0.1540 (0.3713) | 0.0818 (0.4851) |
| | *A. rusticus* | 0.1606 (0.6567) | | |
| Red-crowned tree | *H. ligniperda* | 0.1079 (0.5275) | 0.1178 (0.4275) | |
| | *Shirahoshizo* sp. | 0.1542 (0.5000) | 0.1233 (0.4850) | 0.1108 (0.3046) |
| | *A. rusticus* | 0.1558 (0.5903) | | |
| Dead tree | *H. ligniperda* | 0.0683 (0.5344) | 0.0910 (0.5127) | |
| | *Shirahoshizo* sp. | 0.1151 (0.5515) | 0.1496 (0.4024) | 0.0689 (0.2784) |

Note: The content in parentheses represents the ecological niche similarity.

### 3.4. Coefficients of Competition between the Boring Insects

The competition coefficient between *H. ligniperda* and the other three species of insect was relatively low, and there was no intense competition in host trees of any health status (Table 6). The interspecific competition coefficient between *Cryphalus* sp. and *A. rusticus* was the largest, indicating that the competition between the two was more intense. However, the competition coefficient between these two insects was lower in the dead trees, indicating a weakening competition between them. Combined with variations in the populations of insects at various heights of *P. thunbergii* trees at different health levels, and analysis of the ecological niche similarity and niche overlap index ratio, *A. rusticus* was mainly distributed in the middle and lower parts of the trunk in dead trees, while *Cryphalus* sp. shifted to the middle and upper parts of the trunk, with each occupying a different spatial ecological niche and showing a low degree of overlap in the utilization of resources. The competition coefficient between *H. ligniperda* and *Cryphalus* sp. was lower, mainly because of the differences in the niche occupied by both species; *H. ligniperda* was mainly distributed in the root, while *Cryphalus* sp. was only distributed in the trunk, with the latter moving to the higher part of the tree with the weakening of the tree, and the former moving in the opposite direction.

**Table 6.** Interspecific competition coefficients among boring insects.

| Health Status of Pine Tree | Species | Interspecific Competition Coefficients | | |
|---|---|---|---|---|
| | | *Cryphalus* sp. | *A. rusticus* | *H. ligniperda* |
| | *A. rusticus* | 0.8177 | | |
| Yellow–green tree | *H. ligniperda* | 0.6373 | 0.4942 | |
| | *Shirahoshizo* sp. | 0.4661 | 0.4394 | 0.4914 |
| | *A. rusticus* | 0.7673 | | |
| Red-crowned tree | *H. ligniperda* | 0.5480 | 0.4445 | |
| | *Shirahoshizo* sp. | 0.5460 | 0.4946 | 0.3432 |
| | *A. rusticus* | 0.6866 | | |
| Dead tree | *H. ligniperda* | 0.6260 | 0.5177 | |
| | *Shirahoshizo* sp. | 0.5717 | 0.4128 | 0.2632 |

## 4. Discussion

In terms of temporal niche, *H. ligniperda* and several other insects all feed and cause harm at different times, and there is a certain overlap in temporal resources. Among them, the temporal niche overlap index between *H. ligniperda* and *Cryphalus* sp. is the largest. The temporal niche overlap index is only one aspect of assessing the relationship between species, and the actual competition or coexistence relationship may be affected by many other factors. Therefore, the relationships between *H. ligniperda* and several other insects should be analyzed in combination with spatial distribution.

In this study we determined that *H. ligniperda* affects the roots and the base of the trunk of the host tree [38]. *Cryphalus* sp. prefers feeding on the trunk, while *A. rusticus* and *Shirahoshizo* sp. damage the middle and lower parts of the trunk and the upper part of the roots, with distribution positions overlapping with those of *H. ligniperda* and *Cryphalus* sp.

*H. ligniperda* has a relatively narrow ecological niche width, with low overlap and similarity coefficients in ecological niches compared with several other insect species. This is mainly because it primarily affects the roots and the base of the trunk, occupying a limited space within its host. Due to the difference in spatial distribution and location, though the overlap index of temporal niche with *Cryphalus* sp. is relatively high, the competition between *H. ligniperda* and other insects is not fierce, which may explain the sharp increase in the population of *H. ligniperda* in the dead tree.

*Cryphalus* sp. has the greatest ecological niche width, was the most widely distributed in the trunk, and had a large population therein. Among the insects of *Cryphalus* sp., *Tomicus piniperda* (C. Linnaeus, 1758), *Blastophagus minor* (Hartig, 1834), and *Cryphalus fulvus* (Niisima, 1908) hold a significant numerical advantage and can be found in the trunks of pine trees from 2 to 10 m, more commonly in the 4–10 m range, and they occupy a superior ecological niche in the middle and upper parts of the host tree [39,40]. Therefore, *Cryphalus* sp. had the highest ecological niche overlap value and ecological niche similarity index with *A. rusticus*, and the competition was also more intense. However, in dead *P. thunbergii* trees, the competition coefficient with *A. rusticus* decreased, which may be explained by the fact that the distribution of *Cryphalus* sp. shifted to the higher parts of the dead trees and the density of the insect population differed significantly at different heights of each trunk, coupled with the fact that *A. rusticus* fed inside the xylem during the late larval stage [31,40]. This segregated the feeding site, leading to the co-existence of *Cryphalus* sp. and *A. rusticus*.

Ecotope width, ecotope overlap, and similarity coefficients reflect the degrees of space and resources occupied by the species in a specific geographical area. However, there are limitations in assessing the impact of a species on its host by combining its population and the niche occupied in the host. In this study, although *Cryphalus* sp. dominated in numbers and had the widest ecological niche, its body size is small, with a length of about 2 mm, and its resource utilization was much lower [41,42] than that of *H. ligniperda* and *A. rusticus*.

*A. rusticus* was mainly distributed in the upper 1 m of the root, as well as the middle and lower portions of the trunk, concentrated at the base of the trunk. Previous studies have also shown that *A. rusticus* can affect *P. thunbergii* trees by varying degrees, with the populations of both adults and larvae being mostly concentrated at the base of the trunk, with significant differences from the middle and the top portions of the trunk [43,44]. These findings are consistent with the results of this study. There were significant differences in the populations and distributions of *A. rusticus* in *P. thunbergii* trees with different health conditions, being more abundant in red-crowned and dead trees than in yellow–green trees, and shifting towards the roots and basal 3 m of the trunk with the weakening of the tree. A study by Lu Zhaojun et al. [45] also showed that the population of *A. rusticus* larvae was the greatest in the basal segment of the *P. thunbergii* tree's trunk and decreased upwards, which is consistent with our findings.

During our study, the population of *Shirahoshizo* sp. in *P. thunbergii* trees was generally small, representing only 0.2%–0.3% of the total insect population in the host tree. It was mainly distributed across the basal 3 m of the trunk to the upper 1 m of the root, mostly

concentrated at the base of the trunk. Other studies also show that *Shirahoshizo* sp. are mostly concentrated at the bases of tree trunks, mainly affecting the region below 2 m at the base of the trunk. In this study, this insect was only distributed below 1 m at the base of the trunk, with a relatively small number (5–10 insects). This difference in distribution could be attributed to the dominance of *H. ligniperda* in the root region, which restricted the distribution of *Shirahoshizo* sp.

In line with previous studies, our study showed that different species achieve population coexistence through the allocation and compensation of temporal and spatial resources. For example, Wu Chengxu et al. [46] studied the interspecific relationships and spatiotemporal ecological niches between three *Tomicus* sp. and reported that each of the three species occupied a certain ecological niche on the trunks of Pinus trees. Also, there were differences in temporal and spatial resource utilization, with *B. minor* and *Tomicus yunnanensis* (Kirkendall & Faccoli, 2008) achieving population coexistence competition by allocating and compensating for the temporal and spatial resources. Similarly, Wang Ming et al. [47] explored the spatial ecological niche of *Sirex noctilio* (Fabricius, 1773) and *Sirex nitobei* (Matsumura, 1912), two species of tree wasp that seldom coexist in the same part of the same host, with the former affecting a slightly lower part of the host tree, thereby segregating their spatial ecological niches in order to achieve coexistence.

## 5. Conclusions

In conclusion, *H. ligniperda* and other insects achieve population coexistence through differences in spatial distribution. The population of *Cryphalus* sp. is larger than that of *A. rusticus* and *H. ligniperda*, but its utilization of tree resources is not high. Moreover, although time-based changes in the ecological niche are small, the ecological niche's width increases with the weakening of the host tree, especially in the case of *H. ligniperda* during the later stages of tree damage, during which the borer can occupy a dominant position. Therefore, *H. ligniperda* has the greatest potential for damage among the four insect species.

**Author Contributions:** L.B. and J.T. conceived and designed the experiment. L.B. collected the samples in all periods. L.B. performed the experiment, analyzed the data, and wrote the manuscript. L.B., J.T. and L.R. reviewed and edited the manuscript. C.W. and K.Z. guided the experimental methods and provided financial support for the project leading to this publication. All authors have read and agreed to the published version of the manuscript.

**Funding:** This research was supported by the National Key Research and Development Program of China (2021YFD1400300), and the Yantai City Science and Technology Innovation Development Plan Basic Research Project (2023JCYJ104).

**Data Availability Statement:** The database is available upon request to the correspondence author.

**Acknowledgments:** We would like to express our special thanks to Jianlin Wang, Zhengyi Li, Ling Cheng, Xuesong Chen, Zhiqian Chen, and Imaging Zhuo Zong for their help and support. Additionally, we are thankful for the concern and support from Changlin Wang and others towards our experiment, as well as the guidance and help from the senior members of our research group.

**Conflicts of Interest:** The authors declare that this research was conducted in the absence of any commercial or financial relationships that could be construed as potential conflicts of interest.

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
