# Peer review of "Ecological Niche Studies on Hylurgus ligniperda and Its Co-Host Stem-Boring Insects"

_forests, doi:10.3390/f15050792_

Round 1

Reviewer 1 Report

Comments and Suggestions for Authors

I am grateful to the authors for an interesting manuscript and a complex experiment carried out in the field.

The study of secretive stem pests is accompanied by complex methodological and methodological work, but the authors have successfully coped with this task.

I have an important remark:

1. I would like to see the age of the model trees when describing them. Is it possible to add information on the age of model trees with different degrees of desiccation to the Materials and Methods section?

2. Have data on larval biomass of the studied beetle species been obtained? What is the biomass ratio of these species?

Author Response

亲爱的审稿人:

Greetings!

I have completed the revision of the manuscript. I have carefully considered the valuable comments and suggestions made by both reviewers and have carried out word-by-word and sentence-by-sentence proofreading and modifications, as well as addressed the questions you raised. Please see the attachment for details. Should you find any other issues or require further modifications during your review, please do not hesitate to inform me. I would be more than happy to make further adjustments and improvements.

Thank you once again for your patience, guidance, and assistance. I look forward to your response.

Reviewer 2 Report

Comments and Suggestions for Authors

The topic of the study is very interesting, especially because this species is new to the Chinese fauna. The introduction is satisfactory, while Material and methods section has some serious shortcomings. I've noted all the serious changes that should be made to this chapter, and some minor changes to the sections before it. Without adding the suggested information to the Material and methods - which development stages of the insects were collected, how were they collected, how were they identified, how many trees were used for the statistical analysis (only 5 per the damage degree?), what statistical analysis and how was it conducted, was the sample analyzed per tree, per section, per plot, etc. , and the first part of the results - number of collected individuals, etc. All other suggestions are in the attached file. Without improving these parts, the rest of the manuscript does not make sense reading. Try to improve these things and then I would gladly continue the review.

Comments on the Quality of English Language

The English language is of moderate quality. However, some words and phrases are just wrong. Some text in Chinese is left on the figures. It needs improving.

Author Response

Dear Reviewers,

Greetings!

I have completed the revision of the manuscript. I have carefully considered the valuable comments and suggestions made by both reviewers and have carried out word-by-word and sentence-by-sentence proofreading and modifications, as well as addressed the questions you raised. Please see the attachment for details. Should you find any other issues or require further modifications during your review, please do not hesitate to inform me. I would be more than happy to make further adjustments and improvements.

Thank you once again for your patience, guidance, and assistance. I look forward to your response.

Round 2

Reviewer 2 Report

Comments and Suggestions for Authors

Dear authors,

Although you have improved the manuscript significantly, there are more points that need addressing. 

Remove the unnecessary text that is only burdening the manuscript. 

Change the Latin names of the species according to the suggestions in the attached document.

Add the hypotheses, or the research goals as i requested in the previous review. I've added some potential suggestions in the attached document.

Some of the methods of analysis are missing from the material and methods section. I.e. you don't mention omnibus test, you don't mention anything about which predictors were used for the GLM, or which method you used for the temporal niche analysis. There is no need to mention that you sorted data in excel. The materials and methods section should be detailed enough to enable a researcher to replicate your experiment. Add the data on the insect identification in the materials section, i.e. at least just cite the literature used for identification.

There are many technical mistakes that I didn't review in detail - double spaces, capital instead of small letters, errors in the figures, etc.

The figure and table titles should be self-sufficient in describing what is in the table - add more details to them.

As the insects were not separated by their development phase, the whole temporal niche chapter is questionable, and it should be explained why it was done as such, and not by the development phase. I am sure that you would have much more interesting results if you separated them during the analysis.

The rest of the point by point suggestions are listed in the attached document.

Comments on the Quality of English Language

The manuscript is still to wordy, although you removed many redundant words. There are missuses of the words such as Gall, or I am mistaking, as I don't understand what you wanted to say as I asked in the previous review

Author Response

Dear Reviewer,

I am grateful for the time and effort you have invested in reviewing my manuscript. I appreciate your valuable feedback and comments, which have helped me improve the quality of my work.

I have carefully considered each of your comments and suggestions, and I am responding to them below.

Once again, I appreciate your comments and suggestions. They have been instrumental in improving the quality of my work. I am committed to addressing your concerns and ensuring that my manuscript meets the highest standards of academic publishing.Please refer to the attachment for the specific reply content.

Thank you for your valuable feedback.

Best regards,

Lihong Bi
